# Designing chemical systems for precision deuteration of medicinal building blocks

Jonathan D. Dabbs [1], Caleb C. Taylor [1], Martin S. Holdren[1], Sarah E. Brewster[1], Brian T. Quillin[1,2], Alvin Q. Meng [1], Diane A. Dickie [1], Brooks H. Pate [1] ✉ & W. Dean Harman [1] ✉

Methods are lacking that can prepare deuterium-enriched building blocks, in the full range of deuterium substitution patterns at the isotopic purity levels demanded by pharmaceutical use. To that end, this work explores the regio- and stereoselective deuteration of tetrahydropyridine (THP), which is an attractive target for study due to the wide prevalence of piperidines in drugs. A series of $d_0-d_8$ tetrahydropyridine isotopomers were synthesized by the stepwise treatment of a tungsten-complexed pyridinium salt with $H^-/D^-$ and $H^+/D^+$. The resulting decomplexed THP isotopomers and isotopologues were analyzed via molecular rotational resonance (MRR) spectroscopy, a highly sensitive technique that distinguishes isotopomers and isotopologues by their unique moments of inertia. In order to demonstrate the medicinal relevance of this approach, eight unique deuterated isotopologues of *erythro*-methylphenidate were also prepared.

Over the past decade, there has been a resurgence in the use of deuterium for the active pharmaceutical ingredients (APIs) of medicines[1,2]. New methods for late-stage hydrogen-deuterium exchange provide rapid access to various isotopic variants of existing drugs but suffer from broad distributions of isotopologues (differing in number of deuteriums) and isotopomers (differing in position of the deuterium)[3–7]. Emphasis has been placed increasingly on the development of deuterium-enriched building blocks, not only for the synthesis of known drug derivatives[8] but also for de novo drug design[2]. While access to these elementary moieties enables the preparation of individual isotopomers and isotopologues, most are simple in design and are only available in a limited number of isotopic variations. Methods are lacking that can prepare such building blocks in a full range of deuterium substitution patterns at the isotopic purity levels demanded by pharmaceutical use. To this end, we became interested in preparing deuterated versions of piperidine, which is an attractive target for study due its wide prevalence in drugs[9,10].

Despite years of collective research and considerable financial investment, 90% of drug candidates entering clinical trials fail; much higher percentages never even advance to this phase[11]. One strategy for improving upon these figures is bioisosterism, the process of replacing one portion of a known drug with another closely resembling

it in shape and function, in order to improve the pharmacokinetic profile while retaining the bioactivity of the original compound[12]. A special case of bioisosterism uses the selective placement of deuterium in an active pharmaceutical ingredient (API), leveraging the kinetic isotope effect to improve metabolic stability[1]. But deuterium incorporation can also affect drug performance in more complex ways, including switching metabolic pathways altogether[13] and slowing the epimerization of chiral centers[14]. Recent reports have further suggested that replacing protium with deuterium in drugs might be able to directly alter drug-target interactions[15,16]. Indeed, several deuterated analogs of existing drugs have shown improved performance, and this has spurred research efforts into the precision deuteration of molecules for pharmaceutical chemistry[2]. In 2017, the FDA approved deutetrabenazine (Austedo®) for the treatment of Huntington's disease, in which two $OCH_3$ groups in the parent compound tetrabenazine were replaced with $OCD_3$ groups, improving the pharmacokinetic profile and mitigating side-effects (Fig. 1A)[17]. Furthermore, greater attention has been placed on developing fundamentally new drugs that incorporate deuterium[2]; in 2022, the psoriasis medication deucravacitinib (Sotyktu®) became the first de novo deuterated drug approved by the FDA. Given the complexity of how deuteration might affect drug performance, there is a real need for precision-deuterated samples

[1]Department of Chemistry, University of Virginia, Charlottesville, VA, USA. ✉e-mail: bp2k@virginia.edu; wdh5z@virginia.edu

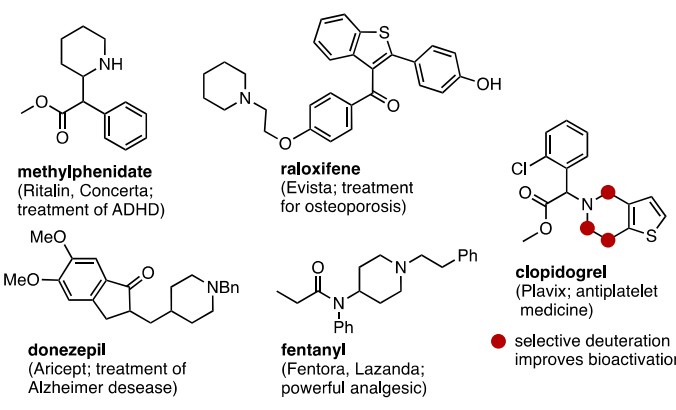

**Fig. 1 | Overview of this study. A** Several new pharmaceuticals contain deuterium. **B** Piperidine rings are common features in medicines. **C** Methods for preparation of deuterium incorporated piperidines from pyridine are generally not regio- or stereoselective. **D** The proposed regio- and stereoselective deuterium incorporation to a pyridine complex and its application to methylphenidate derivatives (ˣH = hydrogen or deuterium). Red indicates positions of deuteration. Orange, blue, purple indicate targeted H/D.

dominated by a single isotopic variant, so that the effects of that particular isotopic substitution may be understood.

The task of obtaining precision-deuterated compounds for medicinal chemistry analysis falls entirely on synthetic chemists since there is no realistic method to separate an isotopic mixture. A recently formed pharmaceutical industry group has summarized these challenges[18]. There is a need for synthetic methods capable of high levels of deuterium incorporation at specific target sites, with minimal isotopic impurities due to over-, under-, or mis-deuteration. Creating new medicines from well-defined deuterated building blocks is one strategy to tightly control the purity and isotopic distribution of potential drug candidates.

Piperidine is the most commonly occurring *N*-heterocycle in FDA-approved small-molecule drugs. Examples include Ritalin®, Plavix®, Lasanda®, and Evista® (Fig. 1B)[9]. Zhu et al. demonstrated that the selective replacement of hydrogen with deuterium in the piperidine ring of the prodrug clopidogrel (Plavix®) resulted in a substantial increase in the active metabolite generated due to a reduction in enzymatic piperidine ring attrition[8]. Their study emphasizes the value of developing new methods for the regio- and stereoselective synthesis of piperidine isotopologues and isotopomers. While various catalytic[3–7] and metal-free[19–21] methods for incorporating deuterium

into pyridine rings have been developed, methods for the selective preparation of deuterated piperidine and tetrahydropyridine (THP) isotopomers are still lacking. In the aforementioned Zhu study[8], deuterium was incorporated into the piperidine core of clopidogrel by assembling the ring via a Mannich condensation with deuterated formaldehyde. This procedure was adapted in the synthesis of a tetra-deuterated analog of the NPY$_5$ receptor SCH 430765[22,23]. Methods have also been developed for H/D exchange at the C2 position of a pyridine or piperidine ring (Fig. 1C)[3,24,25]. Deuterated piperidines and THPs have also been prepared via the reduction of pyridines, either through catalytic reduction of pyridines with deuterium gas (D$_2$)[26,27], by deuterated ammonium formate[28], or other deuteride sources[29–34]. However, all these approaches suffer from poor regioselectivity, poor stereoselectivity, over-deuteration, or extensive overlap of signals in NMR spectra, thereby preventing conclusive analysis of the target and its isotopic impurities.

We envisioned an organometallic approach to the synthesis of THPs[10] and by extension to piperidines, comprising a series of selective deuterium additions to a tungsten-coordinated pyridine. This approach was inspired by our recent experiences with the stepwise reduction of benzene[35]. We anticipated that a similar process could be carried out on the previously reported η$^2$-(*N*-mesyl)pyridinium

tungsten complex **4D** (Fig. 1D)[10]. In this work, a series of $d_0$-$d_8$ tetrahydropyridine isotopomers is synthesized by the stepwise treatment of a tungsten-complexed pyridinium salt with H−/D− and H+/D+. In order to assess the effectiveness of our synthetic methodology, we have employed molecular rotational resonance (MRR) spectroscopy[35], which identifies compounds based on their unique moments of inertia, to precisely determine the isotopic purity of each stereoisotopomer prepared[36,37], and to gain insight into the pathways that lead to side products.

## Results and discussion

### Synthesis of light THP complexes ($d_0$–$d_2$)

The synthetic scheme for creating deuteration patterns on THP is presented in Fig. 2. The pyridine borane complex **4D** was previously synthesized by exchanging pyridine borane onto the tungsten-anisole precursor **1**, thereby forming a mixture of coordination diastereomers, **2D** and **2P** (Fig. 2A)[10]. This mixture can be converted to the pyridinium complexes **3D** and **3P** via diphenylammonium triflate (DPhAT) in acetone and then mesylated via methanesulfonic anhydride in the presence of lutidine to form the $\eta^2$-pyridinium complex **4D**. While the asymmetric nature of the {WTp(NO)(PMe$_3$)} fragment ([W]; Tp = hydridotris(pyrazolyl)borate) gives rise to diastereomers[10], the ratio of the distal (**4D**) and proximal (**4P**) forms can be enriched to >20:1 upon mesylation, owing to the [W] fragment's preference to orient positive charges distal to the PMe$_3$[38]. This system provides good scalability (5 g), consistent purity, and dependably selective incorporation of nucleophiles at its iminium carbon C2 over C4 (isomer ratio (*ir*) > 99:1), thus making it an ideal point of origin for this work.

With a suitable piperidine precursor in hand, our attention turned to incorporating hydrogen and deuterium into the pyridine ring. Compound **4D** was reduced via NaBH$_4$ at −30 °C to form the dihydropyridine (DHP) complex **5** (Fig. 2B), as informed by 2D NMR data (SI). This structure was also obtained via single crystal X-ray diffraction (SC-XRD–Fig. 2C). We observed that this complex has low solubility in methanol, and this property could be utilized to remove impurities through trituration, thereby avoiding chromatography. While the reduction of free sulfonyl pyridinium via NaBH$_4$ results in a mixture of 1,2- and 1,4-DHP products[30], the sulfonyl pyridinium complex **4D** is reduced exclusively to form the $\eta^2$-1,2-DHP **5** (Fig. 2B). A similar reduction was successful with the *N*-acetyl derivative[39]. When this reaction was repeated replacing NaBH$_4$ with its deuterated analog (NaBD$_4$; 99% D), $d_1$-**5** was prepared. **5** and $d_1$-**5** mirror each other spectroscopically in most features except that a ¹H NMR spectrum of $d_1$-**5** shows a broadened singlet integrating to one proton in place of two doublets (each integrating to one proton) in **5**. While the chemical shifts of the diastereotopic methylene resonances in **5** are too similar to determine the stereochemistry of deuteration via NOE interactions, the predicted addition of D− anti to the metal would be confirmed in subsequent experiments (vide infra).

When dissolved in propionitrile and subjected to triflic acid (HOTf), the DHP complexes (**5** or $d_1$-**5**) undergo facile protonation at C6 to form $\eta^2$-allyl complexes[38]. *N*-sulfonyl enamines are ambiphilic species capable of undergoing both electrophilic and nucleophilic additions at the β position[40]. However, in this case, protonation is observed exclusively at the α position (C6) owing to a strong π-backbonding interaction[10]. The resulting compounds, **6** and 2-$d_1$-**6** are two rapidly interconverting $\eta^2$-allyl conformers **6P** and **6D**, of which there is a thermodynamic preference for the distal form, **6D** (Fig. 2B)[38]. This assignment is supported via NOE interactions between the C6 methylene protons and the PMe$_3$ ligand, as has been observed with similar systems[10,39,41]. An $\eta^2$-1,2,3,6-THP complex (**7**) is formed when NaBH$_4$ is added to C3 of **6** or 2-$d_1$-**6**. The structure of **7** was confirmed via SC-XRD (Fig. 2C). ¹H NMR spectra indicate that this material is formed free of its purported $\eta^2$-1,2,5,6-THP isomer that would have resulted from the hydride addition occurring at C5 of the allyl complex

(**6P**). We note that in previous work when the C2 carbon contains a substituent other than hydrogen, C5 additions occur preferentially to form $\eta^2$-1,2,5,6-THP complexes[10]. Repeating this reaction sequence with 2-$d_1$-**6** results in 2-$d_1$-**7**. Deuterium was also incorporated using NaBD$_4$ for the second hydride addition, resulting in the THP isotopologues 3-$d_1$-**7** and *cis*-2,3-$d_2$-**7** respectively. Initially, the preparation of the THP complex **7** was attempted directly from the DHP complex **5** as a one-pot reaction, combining protonation and hydride addition steps. However, this procedure resulted in varying amounts of an unknown impurity. The extent of the impurity generated directly correlated with the amount of excess triflic acid used. We postulated that the targeted THP complex **7** undergoes nitrogen protonation, which induces a ring-opening to form an allyl complex (**8**; Supplementary Information), which then undergoes addition of a hydride to form an $\eta^2$-(pent-4-ene-1-amine) complex (**9**; Supplementary Information). This side reaction was suppressed by isolating the allyl complexes (**6**) prior to the second hydride/deuteride addition.

An additional opportunity to introduce deuterium is through D+ addition to DHP complexes **5** or $d_1$-**5** (Fig. 2). In an earlier report from our labs, Harrison *et al.* observed that an analogous $\eta^2$-(1-acetyl-2-ethyl-1,2-dihydropyridine) complex undergoes D+ addition anti to the metal[39]. Unfortunately, efforts to stereoselectively deuterate **5** or $d_1$-**5** were largely unsuccessful, resulting in only ~20% deuterium incorporation at the C6 position. This suggested that, like the $\eta^2$-cyclohexadiene analog, an unusually large deuterium kinetic isotope effect (DKIE; $k_H/k_D$) may be responsible for trace amounts of fortuitous H+ successfully competing against the large excess of D+[35].

### Synthesis of heavy THP complexes ($d_5$–$d_8$)

We reasoned that repeating the stepwise reduction sequence in Fig. 2. starting from pyridine-$d_5$ could give us access to the heavy THP isotopologues ($d_5$–$d_8$) -**14**. After conversion of pyridine-$d_5$ to the BH$_3$ analog (50 g; **10**)[42], an isolable mixture of $d_5$-**2P** and $d_5$-**2D** was prepared from **1** (Fig. 2D). The ¹H NMR spectrum of $d_5$-**2** showed each pyridine ring position maintained ≥95% deuterium incorporation. The borane group was then removed as before to form a mixture of $d_5$-**3D/P**, which was then elaborated to $d_5$-**4D**. Similar to that observed for the proteated complex **4D**, the deuterated analog $d_5$-**4D** readily incorporates both hydride and deuteride to form the DHP complex isotopologues $d_5$-**5** and $d_6$-**5** respectively (Fig. 2E). When the DHP complex $d_6$-**5** was protonated using the same protocol as **5** (1 eq triflic acid in propionitrile), protium incorporation in the target allyl complex was observed solely at C6 (>95%; 6H-$d_6$-**6**). However, the ratio of syn/anti protonation ratio was only 7:3 according to ¹H NMR peak integrations. The *syn* and *anti* protons of the C6 diastereotopic methylene group were distinguished for the $d_0$-**6** isotopologue via a strong NOE interaction between the PMe$_3$ ligand and the syn proton, which was not present for the anti proton. A variety of conditions were examined to improve the stereochemical purity for this step. Altering the temperature had no noticeable effect, and weaker acid sources resulted in poorer diastereoselectivity. Increasing the equivalents of acid used resulted in increased protium incorporation at both the syn and anti positions on C6, but also at C2. Acetonitrile gave the optimal syn/anti ratio of 9:1. In analogous fashion, *trans-2H,4H-$d_5$*-**6** could be prepared from *2H-$d_5$*-**5** with a dr of 9:1. Incorporating H+ syn relative to [W] has precedent with the protonation of the $\eta^2$-$d_6$-benzene complex[35]. In that study, computations indicated that ring protonation may be occurring via the nitrosyl ligand.

Following the synthesis of a *trans-2H,4H-$d_5$*-**6**, and 6H-$d_6$-**6**, NaBH$_4$ or NaBD$_4$ was added to each of the allyl complexes to render $d_5$-, $d_6$-, and $d_7$- isotopologues of **7** (Fig. 2E), where the $d_6$ isotopologue was prepared as two different isotopomers (*trans-2H,6H-$d_6$*-**7** and *trans-2H,3H-$d_6$*-**7**). Similar to the light isotopologues, the hydride (deuteride) additions occur distal to the PMe$_3$ ligand at C3, exclusively anti to the metal. In order to access a fully deuterated $d_8$-**7** complex, we sought a

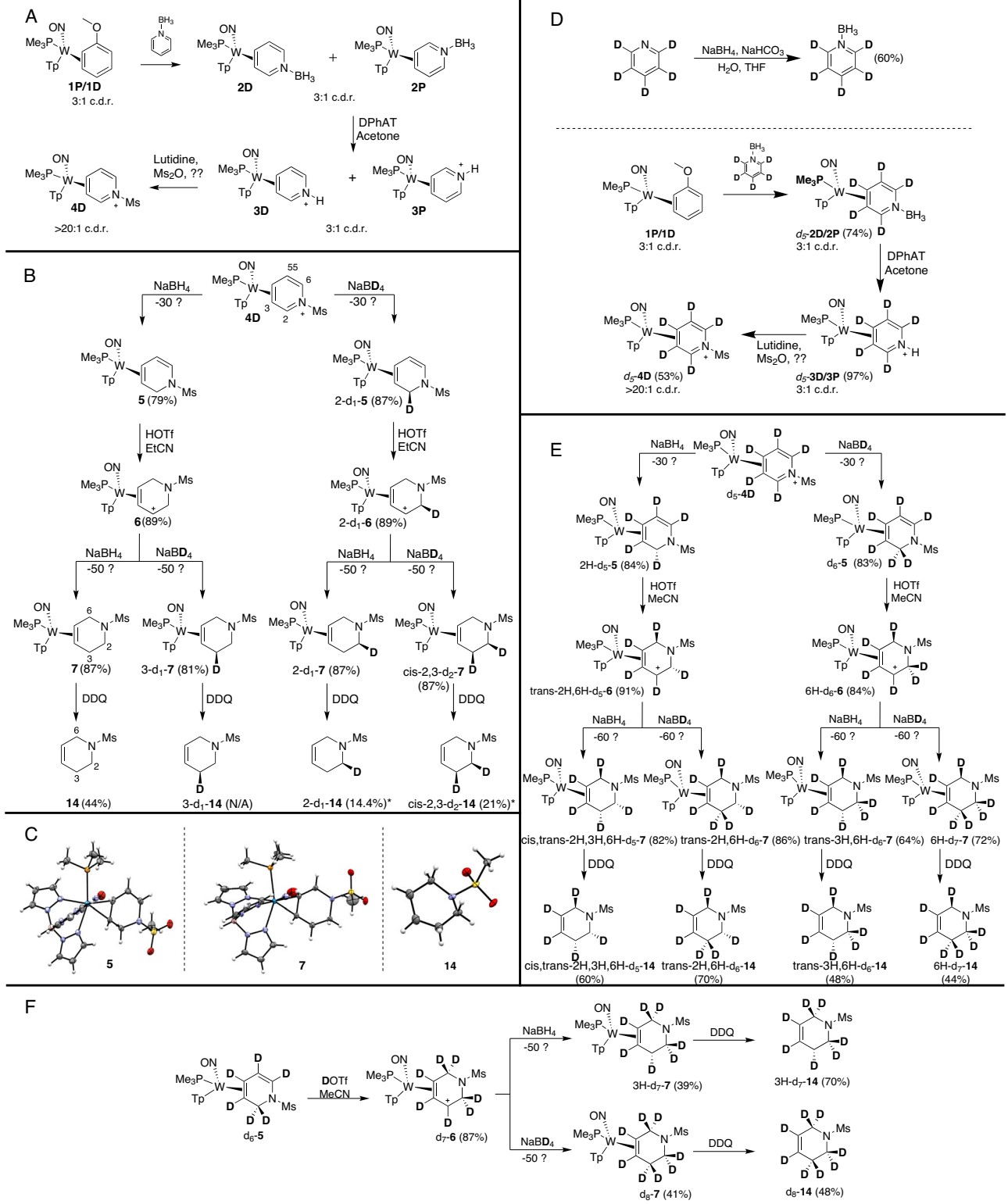

**Fig. 2 | Syntheses of various isotopomers of a mesylated THP. A** The synthesis of a mesylated pyridinium complex. **B** The synthesis of $d_0$, $d_1$, and $d_2$ THP isotopologues from **4D**. **C** SC-XRD ORTEP/ellipsoid diagrams of **5**, **7**, and **14**. **D** Preparation of the $d_5$-**4D** precursor. **E** Synthesis of $d_5$-$d_7$-**14** complexes from $d_5$-**4D**. **F** The $d_6$-**5** complex is elaborated into THP $d_8$-**14**. Compound names have been abbreviated as follows for light isotopologues derived from R hand of the metal: 2-$d_1$-**14** = rel-(2S)-$d_1$-**14**; 3-$d_1$-**14** = rel-(3S)-$d_1$-**14**; cis-2,3-$d_2$ −**14** = rel-(2S, 3S)-$d_2$-**14**. For heavy isotopologues, an additional abbreviation is helpful: $d_5$-**14** = cis, trans-2H, 3H, 6H-$d_5$ −**14** = rel-(2R,3R,6R)-$d_5$-**14**. * = Two Step Yield.

$d_7$-**6** allyl complex $d_7$-**6** by incorporating $D^+$ into $d_6$-**5** (DOTf in acetonitrile; Fig. 2F). While we were unsuccessful in this endeavor with **5** and $d_1$-**5** (*vide supra*), ~80% deuterium incorporation was achieved for

the $d_6$-**5** reaction with ~ 10: 1 *syn:anti* stereoselectivity. Deuteride was subsequently added to $d_7$-**6**, resulting in $d_8$-**7**. Alternatively, the addition of hydride to $d_7$-**6** resulted in the second $d_7$ isotopomer, *3H-$d_7$-**7**.

## Oxidative demetallation of THPs

Liberation of 1,2,3,6-THP organic ligands readily occurs when the corresponding complex is exposed to a chemical oxidant. The oxidation of W(0) to W(I) or W(II) greatly diminishes its backbonding capability, thus weakening the THP-W bond and allowing the dissociation of the organic. 2,3-dichloro-5,6-dicyano-1,4-benzoquinone (DDQ) dissolved in acetone was suitably oxidizing to effect liberation of the THP. To ensure complete decomplexation, an excess of DDQ (4 equivalents) was used. After four hours of stirring, (N-mesyl)-1,2,3,6-THP (14) was cleanly eluted off a basic alumina column (Fig. 2B/E/F). This process was generalized for the isolation of eight different isotopologues of 14 on a 100 mg scale.

## MRR analysis of THPs

In virtually all reactions involving deuterium, numerous isotopic impurities can be produced. To understand the mechanistic origin of isotopic impurities, it is useful to determine the sample composition through quantitative analysis of all chemically distinct deuterated versions of the analyte present in the reaction mixture. While $^1$H NMR spectroscopy can be a useful analytical tool for determining the site and approximate amount of deuterium incorporation in a single compound, quantitative analysis becomes a substantial challenge with mixtures of isotopologues and isotopomers. Since multiple distinct isotopic variants contribute to the proton resonances, it is not possible to determine the sample composition in terms of chemically distinct deuterated versions of the analyte from the $^1$H NMR spectrum. In general, mass spectrometry cannot differentiate isotopomers and, therefore, it is typically used to determine the isotopologue distribution. MRR spectroscopy provides a method to determine the sample composition in terms of the chemically distinct isotopic variants[37]. The rotational spectrum is determined mainly by the moments-of-inertia of the molecule so that both the masses of the nuclei and their positions in the molecular structure define the spectrum[36]. An important feature of the technique is that the spectral pattern for each chemically distinct isotopic variant can be predicted to high accuracy from an equilibrium geometry obtained from quantum chemistry. This approach allows for high-confidence, library-free species identifications. The spectral resolution of MRR instruments is exceptionally high, thus enabling direct analysis of complex mixtures, including the identification of low-level isotopic impurities since spectral overlap is not a significant issue[36]. Other molecular spectroscopy methods, such as infrared or Raman vibrational spectroscopy, do not offer these features limiting their use in the analysis of isotopic impurities of deuterated molecules. In this work, isotopic impurities present down to 0.3% relative to the major species can be identified and quantified in a measurement using approximately 10 mg of sample. For this study, each isotopologue of 14 was analyzed via MRR to confirm the regio- and stereochemistry of the target compound as well as to identify and quantify any impurities present (Fig. 3B). In the analysis, the presence of all under-deuteration (impurities with one fewer deuterium incorporated relative to the target), mis-deuteration (an isotopomer of the target), and over-deuteration variations (with one more deuterium than the target) were checked. A total of 36 variations were calculated, treating the alkene positions as predetermined by the choice of originating pyridine. In this way, it could be shown that specific isotopic impurities are not present at the detection limit and this information can be useful for elucidating the reaction mechanisms that lead to isotopic impurities. The specific deuteration isomers checked for each target reaction product are listed in the Supplementary Information. In these summaries of the isotopic composition of the sample, a value of zero indicates that the isotopic variant could not be detected in the sample.

Whereas NMR data can only provide information about the summed total of isotopic variants in a mixture, MRR gives unique access to the granular details of the mixture composition. This is highlighted when the MRR sample composition analysis was compared to $^1$H NMR data from both the tungsten-complexed and free organic THPs. The integrations of the protons on C2, C3, and C6, the sites of additions, were analyzed in the $^1$H NMR spectra of the highly deuterated ($d_5$–$d_8$) samples and were then compared to expected integrations calculated based on the fraction of each MRR-identified compound present (Fig. 3). For NMR measurements of the complexed THP, the chiral transition metal complex gives resolved diastereomeric resonances for protons at C2 and C6. The integration at these separate resonance positions can be added to give the integration expected for free organic THP. For the MRR measurement, the total amount of deuterium incorporated at C2, C3, and C6 can be calculated from the percentage of each isotopic variant present in the sample weighted by the number of deuterium substitutions at each carbon in that specific molecule[37]. This result can be compared directly to the NMR integrations of the free THP. There are two important points: first, the NMR integrations of the free THP closely match the MRR integrations calculated from the measured sample composition. The 3H-$d_7$-14 and trans-3H,6H-$d_6$-14 samples notably differed between the two techniques by 8% and 9% respectively. The rest of the samples all fell within a 5% range, demonstrating a general agreement between these two analysis methods on the overall composition of each sample. Second, there is also good agreement between the total integration at C2, C3, and C6 for the metal complexed THP and the free THP. This result supports the hypothesis that the deuterium patterns are not modified by the oxidative demetallation of THP, and therefore, the MRR sample composition reflects the composition of THP on the metal after the deuteration reactions are complete, but prior to decomplexation.

## Mechanistic insights

MRR sample analysis revealed several recurring impurity pathways and a detailed mechanistic analysis is provided in the SI for each isotopomer and isotopologue synthesized. The case of cis-2H,3H-$d_2$-14 is provided in Fig. 4 as an example. Given that all these samples were run under similar reaction conditions, generalizations can be made as to which mechanisms are operative: face-flips at the DHP stage (5) and 1,2-hydride shifts at the allyl stage (6) can be ruled out, as can hydride addition syn to the metal in Step 1 and Step 3 (Fig. 4). Under-deuteration impurities ($d_{n-1}$-14) were observed at each deuterium incorporation step (1st deuteride, deuteron, 2nd deuteride (Fig. 4, purple boxes). These were attributed to residual impurities in the system sourced from the reagents themselves, or proton-exchanging sources (e.g., water) either in the solvent or on the glassware. While rigorous drying of solvents and glassware minimized these pathways, they could not be removed altogether. In the case of cis-2,3-$d_2$-14, performing deuteride reactions in CD$_3$OD brought the 2-$d_1$ and 3-$d_1$ impurities from a collective 20.6% to 4.8%. The greatest source of impurities was associated with the protonation of the DHP complexes 5 at C6, which in the case of D$^+$ resulted in significant under-deuteration (see SI; e.g., $d_8$-14; 18.8%). This is likely due to a large deuterium kinetic isotope effect (DKIE) analogous to that previously observed for H$^+$/D$^+$ addition to tungsten $\eta^2$-diene complexes (DKIE ≈ 37 at −30 °C)[35], or H/D isotope exchange at the acidic 2 and 6 positions of the allyl complexes $d_x$-5. In contrast, addition of D$^-$ or H$^-$ in both reduction steps occurred with deuterium yields at or above 95% for all samples.

Mis-deuterations occurred when hydride/deuteride addition took place at C5 (proximal) instead of C3 (distal) on the $\eta^2$-allyl complexes $d_x$-6 (see SI)[10,38]. While 2D NMR analyses of the resulting compounds $d_x$-7 indicate that addition is heavily favored at C3, C5 addition accounts for minor impurities detected via MRR. The ratio of C3 to C5 addition products is found to be approximately 19:1 across various species (see analyses of: cis-2,3-$d_2$-14, cis,trans-2H,3H,6H-$d_5$-14, trans-3H,6H-$d_6$-14, 6H-$d_7$-14). A brief study conducted to determine the temperature dependence of the regioselectivity resulted in no improvement at colder temperatures and increased formation of the pent-4-ene-1-amine complex 9 at −30 °C and above. Furthermore, mis-deuteration can occur during the synthesis of $d_5$-$d_7$ THPs due to protonation at C6 anti to the [W] fragment. Consistent with $^1$H NMR observations, syn addition was favored ~9:1 over anti-addition, and this is supported by MRR data (vide supra).

**Fig. 3 | Analysis of deuterium isotopomers. A** Detailed composition of each sample mixture as determined by MRR. Samples shown include the top two mis-deuterated and top two under-deuterated ($d_{n-1}$) impurities detected for the overall sample (all detectable isotopomer impurities reported in SI). **B** A quantitative comparison of NMR (integration of unobstructed peaks for both the complex (**7**) and the organic THP (**14**)) and MRR data of the free organic (**14**), that shows good agreement between the two techniques (compare numbers within each grey box).

| | cis,trans-2H,3H,6H-d_5-**14** | trans-3H,6H-d_6-**14** | trans-2H,6H-d_6-**14** | 6H-d_7-**14** | 3H-d_7-**14** | d_8-**14** |
|---|---|---|---|---|---|---|
| **7**: NMR H2-syn | 0.06 | 0.05 | 0.05 | 0.08 | 0.02 | 0.03 |
| **7**: NMR H2-anti | 1.01 | 0.07 | 1.01 | 0.03 | 0.06 | 0.10 |
| **7: NMR H2 sum** | 1.07 | 0.12 | 1.06 | 0.11 | 0.08 | 0.13 |
| **14: NMR H2** | 1.03 | 0.10 | 1.01 | 0.10 | 0.07 | 0.05 |
| **14: MRR H2** | 1.02 | 0.11 | 1.02 | 0.10 | 0.07 | 0.05 |
| **7: NMR H3 sum** | 1.09 | 0.98 | 0.09 | 0.10 | 1.02 | 0.10 |
| **14: NMR H3** | 1.02 | 0.92 | 0.04 | 0.05 | 0.94 | 0.05 |
| **14: MRR H3** | 0.99 | 1.01 | 0.03 | 0.05 | 0.82 | 0.02 |
| **7: NMR H6-syn** | 0.94 | 0.90 | 0.93 | 0.91 | 0.14 | 0.12 |
| **7: NMR H6-anti** | 0.09 | 0.15 | 0.08 | 0.16 | 0.08 | 0.07 |
| **7: NMR H6 sum** | 1.03 | 1.05 | 1.01 | 1.07 | 0.22 | 0.19 |
| **14: NMR H6** | 1.00 | 0.97 | 1.00 | 0.97 | 0.21 | 0.17 |
| **14: MRR H6** | 1.00 | 0.97 | 0.99 | 0.96 | 0.24 | 0.21 |

In addition to providing highly detailed analyses of THP mixtures, MRR proved useful in optimizing sample syntheses. As an example, *cis-2,3-d_2*-**14** was originally synthesized with the final step using $d_4$-methanol as the solvent (Fig. 4). MRR shows a combined 13% over-deuteration. These impurities purportedly develop from the transient deprotonation of the allyl $d_1$-**6** back to $d_1$-**5** followed by a D⁺ addition to C6, thereby forming a mixture of $d_2$-**6** allyl complexes. These impurities then undergo deuteride addition to form the over-deuterated *cis,cis*-2,3,6-$d_3$-**14** isotopologue impurity (Fig. 4 pink box). A modified procedure replaced $d_4$-methanol with dry THF, which resulted in a mixture of products that featured less of the target (75%) but a drastic

reduction of over-deuteration (0.3%). This came at the price of increasing the under-deuteration products from 4% to 21%.

## Synthesis of methylphenidate isotopologues and isotopomers

After synthesizing and optimizing a library of THP isotopologues and isotopomers, we next sought to demonstrate the utility of this selective deuteration method in the context of a biologically active piperidine drug. Previous studies from our group have demonstrated that *N*-mesyl analogs of methylphenidate (MPH; Ritalin®) are accessible from the pyridinium complex **4D**[10,43]. The major metabolite of MPH is ritalinic acid[44], a process catalyzed by carboxylesterase and unlikely to be affected by deuterium

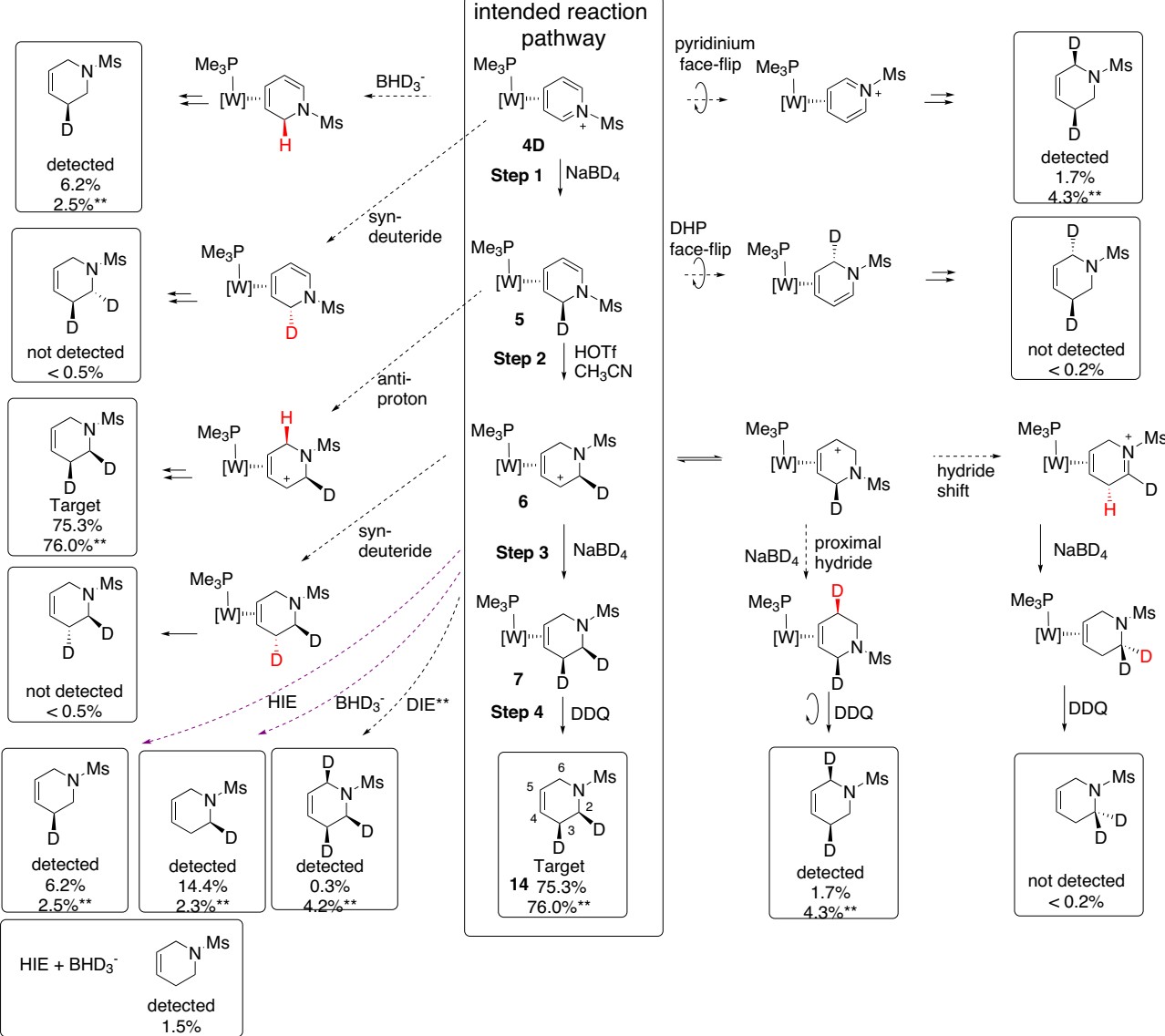

**Fig. 4 | Mechanistic considerations accounting for various isotopologue and isotopomer impurities for the sample *cis-2H,3H-$d_2$-14*.** BHD$_3$: impurity in BH$_4^-$ generated in CD$_3$OD. HIE = exchange hydrogen for deuterium; DIE = exchange deuterium for hydrogen. **Step 3 runs in CD$_3$OD (otherwise in THF). Similar analyses can be found in SI for all other samples of $d_n$-14. Red indicates incorporated H/D.

incorporation into the piperidine ring. However, as mentioned earlier, deuterium incorporation can influence drug-target interactions and mechanism of action[2], which for MPH remains largely unknown. MPH also contains two stereocenters which would provide meaningful stereochemical information regarding the incorporation of deuterium.

Hence, in the presence of methyl α-phenyl bromoacetate (MAPBA), Zn[0], and THF, **4D** will stereoselectively incorporate the methyl phenylacetate moiety at C2 to generate the DHP complex **11** (Fig. 5A). Correspondingly, $d_5$-**11** can be synthesized using $d_5$-**4D**. As previously reported[10], **11** is readily protonated by triflic acid in propionitrile to form an $\eta^2$-allyl complex (**12**) similar to **6** with the favored carbenium orientation being distal to the PMe$_3$. Unlike **6**, when the reaction conditions used to prepare **6** were applied to $d_5$-**11**, proton incorporation was observed almost exclusively syn to the metal at C6 (~20:1 dr by [1]H NMR) to form the allyl complex $d_5$-**12** (Fig. 5B). Emboldened by the enhanced selectivity demonstrated with H[+] additions for the DHP analog **11**, we then attempted D[+] addition to C6 of **11** (Fig. 5A) Given the difficulties we had selectively adding D[+] to **5** (*vide supra*), a large deuterium reservoir was prepared by using $d_4$-methanol. To offset the acid-leveling of the $d_4$-methanol,

6 eq of DOTf were used. $d_6$-**12** was successfully synthesized with deuterium incorporation >93% at the syn C6 position by [1]H NMR. When these same conditions were applied to the proteated analog **11**, 90% deuterium was incorporated syn relative to the metal at C6 ($d_1$-**12**); however, 65% over-deuteration was observed anti to the metal at C6 (i.e., *6,6-$d_2$*-**12**). A compromise was ultimately reached by replacing $d_4$-methanol with acetonitrile but still using six equivalents of DOTf. This process yielded $d_1$-**12** with ~80% deuterium incorporation syn at C6 and no observable over-deuteration. Adding a hydride to **12** was next carried out with NaCNBH$_3$ at −60 °C to form the THP complex **13** (Fig. 5A). Unlike the reactivity observed for **7**, addition of a nucleophile to **12** occurs at C5 instead of C3, resulting in the highly selective preparation of 1,2,5,6-THP complexes rather than 1,2,3,6-THPs. This putatively occurs due to the presence of the bulky methyl phenylacetate group, which raises the kinetic barrier for addition at C3. A range of 1,2,5,6-THP isotopologues ($d_1$-, $d_2$-, $d_5$-, $d_6$-, $d_7$-**13**) and isotopomers (Fig. 5D) were thus synthesized by performing hydride or deuteride additions to the appropriate isotopologue of **12**. Oxidation by DDQ again resulted in the free organics (**15**; Fig. 5A, B) each on a ~100 mg scale (64%).

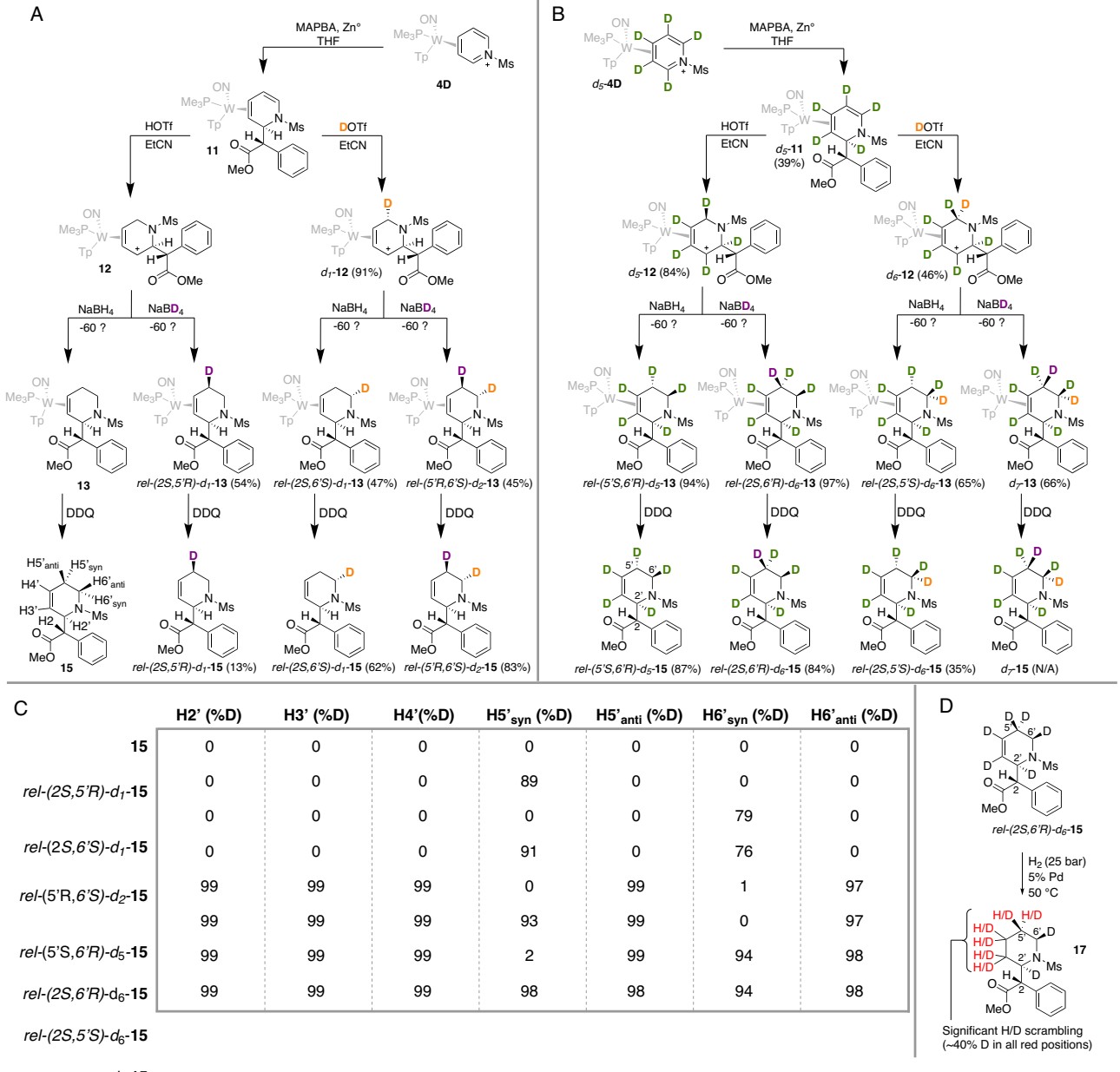

**Fig. 5 | Eight different MPH THP isotopologues/-mers were synthesized. A**: as prepared from **11**. **B**: as prepared from $d_5$-**11**. **C**: quantitative analysis by HNMR for MPH analogs **15** showing the ability to create high purity (green) isotopomers for $d_1$-$d_7$. **D**: hydrogenation to piperidine causes significant scrambling at C3', C4', and C5', but not at C2, C2', and C6. Green, orange, and purple, colors are used to track origin of H or D. Red indicates positions of significant scrambling in hydrogenation step.

## Analysis of MPH derivatives

After establishing general agreement between MRR and $^1$H NMR data for this tungsten-based synthetic method of THPs, the extent of deuterium incorporation for the corresponding MPH organics (**15**) was analyzed via $^1$H NMR (Fig. 5C). The low vapor pressure anticipated for these MPH analogs discouraged us from undertaking an MRR analysis with our current sample holder design. While deuterium incorporation remained an issue for both deuteride and deuteron additions, the protonation/deuteration of C6 (**11**→**12**; Fig. 3A, B) proved more stereoselective than the simple THP analogs. The integrity of the pyridine borane $d_5$-**10** also was demonstrated by the virtually complete loss of signals corresponding to H2', H3', H4', H5', and H6' for the highly deuterated compounds (**15**).

## Catalytic hydrogenation of THPs

Lastly, an isotopomer of $d_6$-**15** was reduced to the fully-saturated piperidine via a catalytic hydrogenation. The conditions for the reduction of **15** was previously reported[10] and similar conditions were utilized in the synthesis of **16**: Using 5% Pd on carbon with 25 bars of $H_2$ at 50 °C, reduction of the THP **14** gave rise to an enamine intermediate (SI) which then gave way to piperidine **16** (confirmed by SC-XRD analysis). When these conditions were repeated in the reduction of *rel-(2S,6'R)-$d_6$*-**15** (i.e., *rel-6'R-2',3',4',5',5',6'-$d_6$***15**) ~ 40% H/D scrambling was observed in **17** for all positions on C3', C4', and C5', thus underscoring the difficulty in selectively generating deuterium-containing scaffolds via Pd catalyzed hydrogenations/deuterations. We note that homogeneous reducing agents could show more promise. While the Chirik catalyst (4-$^t$Bu$^{iPr}$PDI) Mo(CH$_2$SiMe$_3$)$_2$ (PDI = pyridine(diimine)) deuterates benzene and 1,3-

cyclohexadiene with significant scrambling, deuteration of cyclohexene is selective for *cis*-1,2-deuteration to cyclohexane (>98%)[45].

The resurgence in the interest in deuterated compounds in the pharmaceutical industry compels the development of new methods of synthesis for deuterium-containing building blocks available in a wide range of isotopomers. In order to achieve precision deuteration for a family of different patterns, it becomes imperative to have a sequence of synthetic steps that allow for nearly complete deuterium incorporation in each step. Further, in cases where new stereocenters are created by virtue of the replacement of hydrogen for deuterium, these building blocks need to be stereoselectively prepared and analytically verified. Herein we have reported a highly modular methodology for synthesizing a library of piperidine isotopologues and isotopomers ranging from $d_0$–$d_8$. This was accomplished through the regio- and stereoselective additions of deuterides and deuterons to an $\eta^2$-pyridinium complex. In the case of the MPH analogs (**15**), up to four stereocenters were stereoselectively determined by the [W] fragment (*dr* ~ 97), and while not a part of this study, earlier work established that such transformations can be carried out enantioselectively (ee ~ 90)[10]. Although the THP/piperidine isotopic substitution patterns generated via the $d_0$- and $d_5$-pyridine borane systems are not exhaustive, other isotopomers and isotopologues certainly would be accessible by this approach starting from $d_2$- and $d_3$- pyridines (e.g., 2,6- or 3,5-dideuteropyridine; 2,4,6- or 3,4,5-trideuteropyridine)[7,20,46–50]. We note that in applications of deuterated compounds in pharmaceutical science, it is often desirable to have highly deuterated isotopologues to separate the labeled compound from natural abundance isotopic variants of the unlabeled compound when using mass spectrometry. By using commercially available deuterated building blocks as starting materials like the $d_5$-pyridine in this work, it is possible to produce custom deuteration patterns with high total deuterium incorporation while maintaining a sample with high purity of the target species[51]. Furthermore, the tungsten can be utilized for adding substituents, either α (e.g., methylphenidate derivatives) or β to the nitrogen[10], without sacrificing the selectivity of the deuterium incorporations. The combination of selective deuteration and functionalization of the pyridine scaffold significantly expands access to selectively deuterated, functionalized piperidine derivatives.

In addition to reporting a synthetic approach for accessing piperidine isotopologues/-mers via this tungsten complex, we have demonstrated the unique ability of MRR as a tool for analyzing the composition, purity, and reaction mechanisms for these precision-deuterations. While techniques such as mass spectrometry convey the extent of overall deuteration, but with no regard to position, and $^1$H NMR provides often incomplete information of the site and extent of deuterium incorporation, MRR analysis provides a highly quantitative measure of both, thereby providing an optimized synthesis of the targeted THP analogs. The combination of the unique organometallic chemistry offered by tungsten with MRR analysis enables a precise modular method for preparing deuterated functionalized piperidines for medical research.

## Data availability

The crystallographic data generated in this study have been deposited in the Cambridge Crystallographic Data Centre under the accession codes CCDC 2298721-2298724 (**5**, **7**, **14**, **16**). These data can be obtained free of charge from The Cambridge Crystallographic Data Centre via www.ccdc.cam.ac.uk/structures. Other data is available in the Supplementary Information file, including NMR spectra, experimental details, crystallographic information, rotational spectroscopy, and HRMS data. Data supporting the findings of this manuscript are also available from the authors upon request.

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

## Acknowledgements

The authors acknowledge the assistance of Dr. Earl Ashcraft in collecting HRMS data.

## Author contributions

W.D.H., J.D.D., and B.H.P. conceived the project. W.D.H., J.D.D., and B.H.P. designed the experiments. J.D.D., B.T.Q., and C.C.T. prepared samples and collected N.M.R. and H.R.M.S. data. D.A.D. carried out X-ray molecular structure determinations. B.H.P., M.S.H., and S.E.B. performed the molecular rotational resonance (MRR) spectroscopy measurements and developed the methods for isotopic composition analysis.

## Funding

National Institutes of Health (1R01GM132205-01; WDH) (50%) and the National Science Foundation (50%) (CHE-2100345; WDH). Single crystal X-ray diffraction experiments were performed on a diffractometer at the University of Virginia funded by the NSF-MRI grant CHE-2018870. Some NMR data were collected on an instrument funded by an NSF-MRI grant (CHE- 2215062; WDH).

## Competing interests

Brooks Pate has an equity interest in BrightSpec, Inc., which commercializes MRR spectroscopy for analytical chemistry applications. The other authors declare no competing interests.
