## [Peer Review File · Nature Communications]

Designing Chemical Systems for Precision Deuteration of Medicinal Building BlocksREVIEWER COMMENTS

Reviewer #1 (Remarks to the Author):

The work by W. Dean Harman et al. demonstrated a method for the regio- and stereoselective deuteration of tetrahydropyridine (THP) to afford numbers of tetrahydropyridine isotopomers via the stepwise treatment of a tungsten-complexed pyridinium salt with proton and hydride sources. MRR spectroscopy was employed to precisely determine the isotopic purity of each stereoisotopomer prepared. Finally, the pathways that lead to side products were proposed.

Precise incorporation deuterium into target molecular's specific site is very challenging, the authors offered a methodology for achievement of this purpose. From this point of view, the work is of potential utilization in new drugs building. However, there are several points that need to be addressed prior to publication:

- 1) As a general note: ^2H NMR spectrum was necessary for direct locating the position of deuterium. The ^2H NMR spectrum for compound 10 may be needed for determination the purity and positions of deuterium.
- 2) Some NMR SPECTROSCOPY should updated, for example, the signals for ^{13}C NMR for Rel-(6R)-d7-14, d8-14 are too weak to analysis.
- 3) In the main text and SI, some errors occurred for symbol ($^{\circ}\text{C}$) in Table S1-S4, Fig. 2
- 4) What is the yield for every reaction step? This is an important factor for evaluating reaction efficiency and feasibility. At least provide yields for some key compounds in Fig. 2 and Fig. 5.
- 5) Why choose tungsten complex with $\{\text{W}(\text{NO})(\text{PMe}_3)\}$ fragment to activate aromatic rings? Can other tungsten analogous also activate aromatic rings with this π coordination mode. From an application perspective, the simpler and easier it is to synthesize tungsten complex, the more advantageous it is.
- 6) Is the coordination mode between metal tungsten and aromatic ring with π or σ coordination mode? Based on your previous work, sometimes it is depicted as σ coordination, while other times it is depicted as π coordination. What are the valence states of these tungsten complexes? Because different coordination modes have a significant impact on the valence state of the metal center, when coordinating with σ

mode, the metal valence state need +2; π coordination mode does not change the valence state of metal center.

7) Can the reaction be designed as catalytic mode, which can greatly improve the utilization rate of metal tungsten. If not, what perspective should be modified? To increase Lewis acidity? Reduce steric hindrance?

Reviewer #2 (Remarks to the Author):

Comments:

Deuteration reactions have attracted enormous attention owing to the increasing interest in merging deuterium into pharmaceutically relevant molecules. The crucial issue is to develop selective deuteration methodologies to precisely control the labeled position, stereochemistry, and the degree of deuteration because these factors greatly affect the physicochemical and biological properties of drug molecules. In this manuscript, Prof. Harman and co-workers report significant advances including synthetic methodologies and analytical methods in this research area. Based on the previously established strategy involving benzene reductive deuteration (ref. 34), this manuscript describes the reductive deuteration of pyridine derivatives in a regio- and stereoselective manner, thus providing general access to diverse deuterated piperidine isotopologues and isotopomers. Almost a full range of tetrahydropyridine isotopomers were synthesized by the stepwise treatment of a tungsten-complexed pyridinium salt with H-/D- and H+/D+. Moreover, the authors introduced a new analytical method, molecular rotational resonance (MRR), to elucidate the structures of isotopomers and isotopologues by their unique moments of inertia, even in a complex isotopic mixture. Finally, they described mechanistic considerations accounting for various isotopologue and isotopomer impurities. In my opinion, this precision deuteration will find wide applications in related medicinal chemistry because pyridine and piperidine rings are recognized as the top 2 most frequently employed heterocycles in FDA-approved drug molecules. Overall, the manuscript is based on impressive empirical evidence and makes an original contribution. Only minor revisions are needed before it can be published.

Several minor points:

1. How about any substituted (one example is OK) pyridines? The yields and selectivities may not be good, but nonetheless these results should be discussed in the main text or Supporting Information. This is very important for late-stage modifications since pyridine- or piperidine-containing drugs are substituted. While I feel a bit bad about this suggestion after the authors already prepared a long and high-quality SI, I believe it is critically important to demonstrate its late-stage application potential.

2. Reaction yields should be added in the Fig. 2.

3. The related reductive deuteration reactions using a similar strategy via the regio- and stereoselective additions of H-/D- and H+/D+ to metal-bound (hetero)arenes (Angew. Chem. Int. Ed. 2023, 62, e202218961; CCS Chem. 2024, DOI: 10.31635/ccschem.023.202303573) should be cited appropriately.

Reviewer #3 (Remarks to the Author):

The authors have described a substantial amount of novel data on the synthesis and characterization of novel partially deuterated compounds with possible usage in medicinal chemistry and drug development. The reaction pathways have been reported in sufficient detail, and the compounds obtained have been fully and correctly analyzed.

A major technique used in the characterizations reported is Molecular Rotational Resonance. Whereas this technique certainly has a strong added value, it is not clear to me how other types of analysis including for example NMR or vibrational spectroscopy could get in analyzing and thus be able to get similar data. I suggest the authors to comment on this, and to expand the analyses and to really show how far one could go in such analyses by showing results of such an analysis and by clearly identified and describing the pitfalls or problems that can occur.

The MRR methodology and the computation approaches used in analyzing the data obtained are only shortly described in the supporting information, and a remark is made that this is related to the fact that the full analysis will be reported separately in a follow-up paper. A more thorough description of the analyses made and of the problems encountered in structural characterization in my opinion would improve the quality of the paper. It would also help to identify the read added value of MRR, if present, in comparison with other spectroscopic techniques.

Point by point response to reviewer comments

Reviewer #1 (Remarks to the Author):

The work by W. Dean Harman et al. demonstrated a method for the regio- and stereoselective deuteration of tetrahydropyridine (THP) to afford numbers of tetrahydropyridine isotopomers via the stepwise treatment of a tungsten-complexed pyridinium salt with proton and hydride sources. MRR spectroscopy was employed to precisely determine the isotopic purity of each stereoisotopomer prepared. Finally, the pathways that lead to side products were proposed. Precise incorporation deuterium into target molecular's specific site is very challenging, the authors offered a methodology for achievement of this purpose. From this point of view, the work is of potential utilization in new drugs building. However, there are several points that need to be addressed prior to publication:

1) As a general note: 2H NMR spectrum was necessary for direct locating the position of deuterium. The 2H NMR spectrum for compound 10 may be needed for determination the purity and positions of deuterium.

We would respectfully argue that MRR provides far more accurate information about the isotopic purity and precise location of the deuterium atoms than ²H NMR spectroscopy can. While we do not have available a ²H NMR spectrum for **10**, we do have a ²H NMR spectrum for *d*₅-**4D**, which we have now included in the SI (Figure S7 (page S37)). The locations of the peaks correspond to the location of the ligand ring peaks for the ¹H NMR of **4D**, and an NMR of this compound is provided on S30 of reference 10.

2) Some NMR SPECTROSCOPY should updated, for example, the signals for 13C NMR for Rel-(6R)-d7-14, d8-14 are too weak to analysis.

¹³C analyses for these samples are particularly challenging due to the high amount of deuteriums present in both samples. The carbons with deuteriums attached have weaker signals than for the proteated analogs (no NOE enhancement), and short and long range deuterium coupling along with quadrupole broadening renders the spectrum difficult to analyze. To illustrate this point, we call to attention the magnitude of the product peaks with respect to the NMR solvent peak (CD₂Cl₂) in the ¹H spectrum vs the ¹³C spectrum for both samples listed above. In both samples, the ¹H and ¹³C were acquired from the same sample tube. And yet despite the ¹H indicating that both samples are quite concentrated, the ¹³C peaks are still exceedingly small. We would respectfully suggest that this feature further underscores the point that MRR is far more effective for analyzing these samples.

3) In the main text and SI, some errors occurred for symbol (°C) in Table S1-S4, Fig. 2.

We are unable to detect this technical issue in our documents, but will be happy to address this point at the production stage.

4) *What is the yield for every reaction step? This is an important factor for evaluating reaction efficiency and feasibility. At least provide yields for some key compounds in Fig. 2 and Fig. 5.*

Thank you for bringing this to our attention. Yields have been added to Fig. 2-5.

5) *Why choose tungsten complex with {WTP(NO)(PMe₃)} fragment to activate aromatic rings? Can other tungsten analogous also activate aromatic rings with this π coordination mode. From an application perspective, the simpler and easier it is to synthesize tungsten complex, the more advantageous it is.*

The dihapto coordination of aromatic molecules requires highly π -basic metal complexes capable of substantial backbonding. Therefore, 3rd row metals are generally best suited for this. However, the metal cannot be so electron-rich that it undergoes C-H activation. Thus, the metal electron density must be finely tuned (using electrochemistry as a general guide; This optimization has been carried out for Re(I) and Os(II) dearomatization agents as well). Because of this, the [W] fragment is highly sensitive to changes in the electronics brought about by the ligand set. A complete discussion of this organometallic chemistry background can be found in reference 38.

6) *Is the coordination mode between metal tungsten and aromatic ring with π or σ coordination mode? Based on your previous work, sometimes it is depicted as σ coordination, while other times it is depicted as π coordination. What are the valence states of these tungsten complexes? Because different coordination modes have a significant impact on the valence state of the metal center, when coordinating with σ mode, the metal valence state need +2; π coordination mode does not change the valence state of metal center.*

As this referee points out, there are alternative ways of depicting the interaction between metal and ligand in these complexes. The two most conventional methods are: as W(0) complexes bound to the π -bond of an alkene, or as a W(II)-metallocyclopropane complex (see Dewar-Chatt-Duncanson model). The W(0) depiction has the coordinated carbons depicted as sp^2 hybridized while the W(II) depiction has them as sp^3 hybridized). SCXRD data reveals the actual bond length between these carbons to be ~ 1.44 Å, indicating that reality is somewhere in-between. So while neither depiction is incorrect, we have chosen to depict these structures as W(0) π -complexes because this interpretation more closely resembles the chemistry observed. This depiction also provides a consistency with close to two hundred previous reports in the literature.

7) *Can the reaction be designed as catalytic mode, which can greatly improve the utilization rate of metal tungsten. If not, what perspective should be modified? To increase Lewis acidity? Reduce steric hindrance?*

We readily agree that a catalytic approach capable of performing the same chemistry would be advantageous. However, a catalytic approach using [W] or a similarly contrived η^2 -dearomatization is impractical:

Aromatic ligands η^2 -bound to [W] readily dissociate (BDE \sim 20 kcal/mol) due to the strong thermodynamic drive to reestablish aromaticity. However, η^2 -bound nonaromatic ligands, such as the dihydropyridines and tetrahydropyridines observed in this work, are bound very tightly (BDE \sim 40 kcal/mol). The only reliable method for removing the organic ligand is oxidizing the metal, thus diminishing its ability to back-bond. Although oxidative decomplexation could theoretically lend itself to a catalytic process, the [W] fragment decomposes rapidly upon oxidation. In addition, we point out that the success of the technique described herein depends on the ability of the metal to bind a targeted double bond regio- and stereoselectively. These reactions are under strict kinetic control. A modified system in which the metal more easily binds and releases the organic substrate would result in a loss of this selectivity. These issues are addressed in reference 38 and references listed within that review.

Reviewer #2 (Remarks to the Author):

Deuteration reactions have attracted enormous attention owing to the increasing interest in merging deuterium into pharmaceutically relevant molecules. The crucial issue is to develop selective deuteration methodologies to precisely control the labeled position, stereochemistry, and the degree of deuteration because these factors greatly affect the physicochemical and biological properties of drug molecules. In this manuscript, Prof. Harman and co-workers report significant advances including synthetic methodologies and analytical methods in this research area. Based on the previously established strategy involving benzene reductive deuteration (ref. 34), this manuscript describes the reductive deuteration of pyridine derivatives in a regio- and stereoselective manner, thus providing general access to diverse deuterated piperidine isotopologues and isotopomers. Almost a full range of tetrahydropyridine isotopomers were synthesized by the stepwise treatment of a tungsten-complexed pyridinium salt with H-/D- and H+/D+. Moreover, the authors introduced a new analytical method, molecular rotational resonance (MRR), to elucidate the structures of isotopomers and isotopologues by their unique moments of inertia, even in a complex isotopic mixture. Finally, they described mechanistic considerations accounting for various isotopologue and isotopomer impurities. In my opinion, this precision deuteration will find wide applications in related medicinal chemistry because pyridine and piperidine rings are recognized as the top 2 most frequently employed heterocycles in FDA-approved drug molecules. Overall, the manuscript is based on impressive empirical evidence and makes an original contribution. Only minor revisions are needed before it can be published.

Several minor points:

1. How about any substituted (one example is OK) pyridines? The yields and selectivities may not be good, but nonetheless these results should be discussed in the main text or Supporting Information. This is very important for late-stage modifications since pyridine- or piperidine-containing drugs are substituted. While I feel a bit bad about this suggestion after the authors already prepared a long and high-quality SI, I believe it is critically important to demonstrate its late-stage application potential.

Thank you for raising this important point. A true late-stage approach, defined as the binding of pyridine-containing drugs to [W] for the subsequent stepwise reduction, is unlikely to be a generalizable process. This is due to the tendency of the [W] fragment to indiscriminately bind nonaromatic π -bonds (e.g., ester and amide carbonyl groups) and aromatic π -bonds alike. That said, the [W]-pyridinium complexes have been previously shown to enantio-, regio-, and stereoselectively incorporate a broad range of nucleophile types and functional groups (see references 10, 38, 39, 43). These results coupled with the data provided already strongly suggests a broad array of functional groups could be installed that could then be used in tethering the deuterated THP/piperidine to the rest of the drug, such as was demonstrated for MPH derivatives herein.

More to the point, even demonstrating the direct coordination of 2-, 3- or 4-picoline (analogous to what we have done with pyridine) is complicated by multiple competing regio- and stereoisomers. This is not to say that this chemistry could not be accomplished for other pyridine systems, but each variation would require a considerable amount of optimization before it became practical or general. These will be interesting future studies to be sure, but are not achievable on a three- to six-month timeframe.

2. *Reaction yields should be added in the Fig. 2.*

Thank you for bringing this to our attention. Yields have been added to Fig. 2 and Fig. 5.

3. *The related reductive deuteration reactions using a similar strategy via the regio- and stereoselective additions of H-/D- and H+/D+ to metal-bound (hetero)arenes (Angew. Chem. Int. Ed. 2023, 62, e202218961; CCS Chem. 2024, DOI: 10.31635/ccschem.023.202303573) should be cited appropriately.*

Thank you for directing us to these publications. They have been incorporated into the manuscript (references 29 and 30).

Reviewer #3 (Remarks to the Author):

The authors have described a substantial amount of novel data on the synthesis and characterization of novel partially deuterated compounds with possible usage in medicinal chemistry and drug development. The reaction pathways have been reported in sufficient detail, and the compounds obtained have been fully and correctly analyzed.

1) *A major technique used in the characterizations reported is Molecular Rotational Resonance. Whereas this technique certainly has a strong added value, it is not clear to me how other types of analysis including for example NMR or vibrational spectroscopy could get in analyzing and thus be able to get similar data. I suggest the authors to comment on this, and to expand the analyses and to really show how far one could go in such analyses by showing results of such an analysis and by clearly identified and describing the pitfalls or problems that can occur.*

and 3) *It would also help to identify the read added value of MRR, if present, in comparison with other spectroscopic techniques.*

Thank you for raising this issue. A core principle of this manuscript is to introduce MRR analysis as a uniquely effective analytical method for the identification of mixtures of piperidine stereoisotopomers and isotopologues. We respectfully direct attention to Schemes S1-S9 as they outline the precise composition of each deuterated sample as determined by MRR. Nuclear Magnetic Resonance, vibrational analysis, mass spectrometry, nor any other analytical method can provide this level of information. We note that this project is a collaboration between a synthesis group (Harman) and a physical/analytical chemistry group (Pate) where a goal of the work is to test whether MRR spectroscopy meets the needs of synthetic chemists developing precision deuteration methods. Within our collaboration, the honest assessment of MRR spectroscopy from the synthetic chemistry perspective is that no other analytical technique even comes close.

We believe that this excerpt summarizes the point adequately: “While ^1H NMR spectroscopy can be a useful analytical tool for determining the site and approximate amount of deuterium incorporation in a single compound, quantitative analysis becomes a substantial challenge with mixtures of isotopologues and isotopomers. Since multiple distinct isotopic variants contribute to the proton resonances, it is not possible to determine the sample composition in terms of chemically distinct deuterated versions of the analyte from the ^1H NMR spectrum. In general, mass spectrometry cannot differentiate isotopomers and, therefore, it is typically used to determine the isotopologue distribution. MRR spectroscopy provides a method to determine the sample composition in terms of the chemically distinct isotopic variants.”

This reviewer raises the possibility of using IR spectroscopy for the analysis of deuterated molecules. Currently, the manuscript indirectly addresses the advantages of MRR spectroscopy over IR spectroscopy (or electronic spectroscopy):

“An important feature of the technique is that the spectral pattern for each chemically distinct isotopic variant can be predicted to high accuracy from an equilibrium geometry obtained from quantum chemistry. This approach allows for high-confidence, library-free species identifications. The spectral resolution of MRR instruments is exceptionally high, thus enabling direct analysis of complex mixtures since spectral overlap is not a significant issue.”

We are confident that IR spectroscopy does not offer the spectral resolution to analyze a complex sample mixture where spectral overlap would be expected to be extensive. It is difficult to see that IR spectroscopy could be used to identify and quantify low-level impurities in a mixture of deuterated isotopologues and isotopomers. We are reluctant to explicitly make a comparison between MRR and IR spectroscopy in this manuscript

because, to our knowledge, IR spectroscopy is not a currently used technique in this field of analytical chemistry.

We have adjusted the manuscript to read:

“An important feature of the technique is that the spectral pattern for each chemically distinct isotopic variant can be predicted to high accuracy from an equilibrium geometry obtained from quantum chemistry. This approach allows for high-confidence, library-free species identifications. The spectral resolution of MRR instruments is exceptionally high, thus enabling direct analysis of complex mixtures, including the identification of low-level isotopic impurities, since spectral overlap is not a significant issue. Other molecular spectroscopy methods, such as infrared or Raman vibrational spectroscopy, do not offer these features limiting their use in the analysis of isotopic impurities of deuterated molecules.”

2) The MRR methodology and the computation approaches used in analyzing the data obtained are only shortly described in the supporting information, and a remark is made that this is related to the fact that the full analysis will be reported separately in a follow-up paper. A more thorough description of the analyses made and of the problems encountered in structural characterization in my opinion would improve the quality of the paper.

We acknowledge this concern, and appreciate that MRR is not yet an “off-the-shelf” technique than can be easily replicated by synthetic chemists without considerable expertise or collaboration. That said, we have provided recent references (35,36,37) that describe the method, its applications, and its limitations in detail. We draw attention, in particular, to reference 36 (<https://doi.org/10.1002/ansa.202300021>), which provides a detailed overview of the application of MRR to the analysis of isomeric mixtures. In addition, Ref. 37 has a discussion of the use of MRR spectroscopy to identify the isotopologue and isotopomer composition of mixtures of deuterated species – although the isotopic mixtures analyzed in that work are significantly simpler than the ones reported in the current manuscript (<https://doi.org/10.1021/jacs.1c00884>).

We have identified in the manuscript limitations relevant to the current study, which has to do with the vapor pressure of the sample. As suggested, we are currently preparing a manuscript that will provide a detailed discussion of the exact experiments and methods used in the present study. We do not feel it is helpful to duplicate this manuscript in the SI of the current work.